**Subject Category:**
Biology (whole organism)

theoretical biology/ecology/evolution

eco-evolutionary dynamics, social dilemma, spatial dynamics, pattern formation

**Author for correspondence:**
Chaitanya S. Gokhale
e-mail: gokhale@evolbio.mpg.de

# Ecological feedback on diffusion dynamics

## Hye Jin Park[1] and Chaitanya S. Gokhale[2]

[1]Department of Evolutionary Theory, and [2]Research Group for Theoretical Models of Eco-evolutionary Dynamics, Department of Evolutionary Theory, Max Planck Institute for Evolutionary Biology, August Thienemann Street 2, 24306 Plön, Germany

CSG, 0000-0002-5749-3665

Spatial patterns are ubiquitous across different scales of organization in ecological systems. Animal coat pattern, spatial organization of insect colonies and vegetation in arid areas are prominent examples from such diverse ecologies. Typically, pattern formation has been described by reaction–diffusion equations, which consider individuals dispersing between subpopulations of a global pool. This framework applied to public goods game nicely showed the endurance of populations via diffusion and generation of spatial patterns. However, how the spatial characteristics, such as diffusion, are related to the eco-evolutionary process as well as the nature of the feedback from evolution to ecology and vice versa, has been so far neglected. We present a thorough analysis of the ecologically driven evolutionary dynamics in a spatially extended version of ecological public goods games. Furthermore, we show how these evolutionary dynamics feed back into shaping the ecology, thus together determining the fate of the system.

## 1. Introduction

Evolutionary game theory has been successfully used to describe the evolution of types in a population, be it frequencies of alleles in a biological setting or languages in a cultural setting [1,2]. The most widely used application of this theory is in addressing social dilemmas. Social dilemmas result from a collision between the interests of an individual and that of the group to which it belongs [3]. The ubiquity of social dilemmas is evident by its appearance in pertinent issues such as fishery and wildlife management [4] and global climate change [5]. Biologically relevant scenarios such as foraging strategies [6], group hunting behaviour [7,8] and bacterial secretions interpreted as public goods [9] provide these dilemmas a sociobiological setting. The resolution of social dilemmas lies at the heart of achieving a transition in the level of organization, e.g. evolving multicellularity [10] (or the deconstruction of sociality, as in cancer evolution [11]).

Numerous ways of resolving such dilemmas, elegantly captured by public goods games (PGG), have been proposed

[12,13]. One way of resolving PGG is the imposition of spatial structure on the evolving population. Conceptually, classical ideas such as Wright's island model [14], the haystack model [15], contemporary group selection models [16], evolutionary dynamics with structure and many more [17–20], impose a condition limiting encounters between the interacting agents. Spatial dynamics thus has been successful in resolving social dilemmas maintaining a mixture of cooperators and defectors in the long run [21].

Besides stabilizing cooperation, spatial dynamics also results in intricate spatial patterns under eco-evolutionary processes [22]. Ecological dynamics are incorporated by explicitly accounting for the feedback of population densities on the evolutionary processes and vice versa. We deviate from the classical use of diffusion as a 'constant' and investigate an eco-evolutionary feedback on population mobility. Experimental and empirical studies show that dispersal is a property which could be conditioned on a variety of factors, either environmental or a property of the population under question [23] in both plants and animals [24,25]. Including dispersal, it is possible to explain how populations can avoid extinction in a spatially extended selection-diffusion system. This improves our understanding of the ecological aspects of the diffusion process responsible for the spatial patterns. Making diffusion depend on the total density, we put it on an ecological footing and examine its effect on pattern formation. We formulate simple but general density-dependent diffusions and study their effects on pattern formation. First, we employ various density-dependent diffusion formulations and capture the important diffusion properties for forming different patterns by a crude look. After that, as relevant scenarios, we focus on two distinct density-dependent formulations inspired from growing bacterial cultures and human migration. We establish a connection between the diffusion properties and the observed spatial patterns. The details of the diffusion rule, which can differ between species are shown to be crucial in determining the observed patterns.

## 2. Model

### 2.1. Eco-evolutionary dynamics with diffusion

In PGG, cooperators invest a fixed amount $c$ into a common pool. For $m$ such cooperators, this common pool of value $mc$ is then multiplied by a factor $r$. The benefits of this interaction are returned equally to all individuals participating in the game $S$, thus $rmc/S$. While this is the payoff of a defector, $P_D(m) = rmc/S$, a cooperator, having paid the cost, gets $P_C(m) = P_D(m) - c$. The multiplication factor $r$ determines the value of the public good, bounded as $1 < r$ to ensure that mutual cooperation is better than mutual defection.

In order to incorporate population dynamics, (normalized) densities are introduced instead of frequencies of cooperators and defectors. The sum of cooperator and defector densities $u$ and $v$, lies between zero and unity, $0 \leq u + v \leq 1$. The total population density ranges from extinction, $u + v = 0$, to the maximum density, $u + v = 1$. If the density has not reached the maximum, i.e. $w \equiv 1 - u - v > 0$, then the population can still expand.

The actual number of participants, $S$, is sampled according to the total density with the maximum group size $N$. Individuals have a chance to meet another individual with a probability that is proportional to the total density in a well-mixed population. If the population density is small, individuals meet less often and hence form smaller groups. If the density is high then the maximum group size $N$ can be reached. As a consequence, the game-interaction group size $S$ depends on the total density and ranges from 2 to $N$. The lower bound 2 is natural because we need at least two individuals to interact. If there is only one individual, there is no interaction, the game is not played. The average payoffs for defectors and cooperators, $f_D$ and $f_C$, are then the product of the expected payoffs and the probability $p(S; N)$ summed over all possible group sizes $S$. This gives us [26],

$$f_D = \frac{ru}{1-w}\left(1 - \frac{1-w^N}{N(1-w)}\right) \tag{2.1a}$$

and

$$f_C = f_D - 1 - (r-1)w^{N-1} + \frac{r}{N}\frac{1-w^N}{1-w}. \tag{2.1b}$$

We have set the investment cost $c = 1$ (for details see appendix A).

If there is an opportunity for reproduction ($w > 0$), individuals reproduce according to their average payoffs. All individuals are assumed to have the same constant birth and death rates given by $b$ and $d$,

respectively. The change in the densities of cooperators and defectors over time is given by the following extension of the replicator dynamics [27–29],

$$\dot{u} = u[w(f_C + b) - d] \tag{2.2a}$$

and

$$\dot{v} = v[w(f_D + b) - d]. \tag{2.2b}$$

Without migration between subpopulations, the population dynamics at a given position can be analysed separately as an independent population [29].

Transforming variables from cooperator and defector densities $u$ and $v$ to cooperation fraction $f \equiv \frac{u}{u+v}$ and total density $\rho \equiv u + v$ decouples the evolutionary and ecological parameters, $f$ and $\rho$ [29,30]:

$$\dot{f} = (1 - \rho)f(1 - f)F(\rho) \tag{2.3a}$$

and

$$\dot{\rho} = -\rho d + \rho(1 - \rho)[b + f(r - 1)(1 - (1 - \rho)^{N-1})], \tag{2.3b}$$

where $F(\rho) = -1 - (r - 1)(1 - \rho)^{N-1} + (r/N\rho)(1 - (1 - \rho)^N)$. The asymptotic behaviour of the system is determined by the stabilities of the fixed points in the $f$-$\rho$ space. In this manuscript, we only focus on $d > b = 1$ and $N = 8$ wherein defectors cannot survive without cooperators, and the system undergoes a Hopf bifurcation as $r$ varies at a given $d$ [22,26,31]. For small $r$, extinction ($u = v = 0$) is the stable fixed point while for a large $r$ coexistence ($u, v > 0$) becomes stable. Both cooperators as well as defectors die out for a small rate of return from the public good ($r < r_{\text{hopf}}$).

Including spatial dynamics, the stability of the fixed point can change. By forming patterns, cooperators and defectors can coexist even for $r < r_{\text{hopf}}$ [22]. To include spatial dynamics, we envision subpopulations spatially arranged on a two-dimensional lattice. In each patch, the dynamics of the subpopulation is described by equation (2.2), and individuals, cooperators and defectors, randomly move between adjacent patches. There is no game interaction between individuals who live in different patches. By taking the continuum limit of this spatially structured subpopulations, we can get the changes of densities over time,

$$\dot{u} = \nabla \cdot (D_c \nabla u) + u[w(f_C + b) - d] \tag{2.4a}$$

and

$$\dot{v} = \nabla \cdot (D_d \nabla v) + v[w(f_D + b) - d]. \tag{2.4b}$$

The diffusion coefficients $D_c$ and $D_d$ for cooperators and defectors indicate the speeds of their diffusion, respectively. There is no external in- or out-flux at the boundaries. This dynamics with constant diffusion coefficients is the form of the classical activator–inhibitor system [32]: according to the constant ratio of diffusion coefficients $D = D_d/D_c > 1$, various patterns have been observed. With various $r$ and $D$, different dynamical regimes emerge—from homogeneous coexistence to extinction—with chaos between extinction and diffusion-induced coexistence [22] (figure 1).

## 2.2. Ecological feedback on diffusion dynamics

Diffusion dynamics affects extinction of populations and pattern formation. So far, most research has focused on constant diffusion, and eco-evolutionary effects on the diffusion dynamics have not been explored. However, density-dependent diffusion is observed across scales of organization from microbial systems to human societies [34–38]. The density-dependent diffusion coefficients may have eco-evolutionary components such as $f$ and $\rho$. We examine the effect of this eco-evolutionary diffusion dynamics on pattern formation.

For the sake of simplicity, we assume a fixed diffusion coefficient for cooperators and develop the defector's density-dependent diffusion. These differential mobilities are empirically motivated. Within a population, different types of individuals can show different mobility, for example, as in the aphid and planthopper populations [39–44]. Examples show that when defectors are moving faster than cooperators, it is possible for the population to survive harsh environments [45,46]. It will thus be interesting to focus on $D > 1$. The defector's diffusion coefficient may be written as

$$D_d(f, \rho) = D_c\left[1 + \sigma\frac{g(f, \rho)}{\max_{f,\rho \in [0:1]} g(f, \rho)}\right]. \tag{2.5}$$

The function $g(f, \rho)$ encodes diffusion behaviour and is normalized by the maximum value to bind the range of density dependence from zero to unity. The parameter $\sigma$ acts as the intensity of density

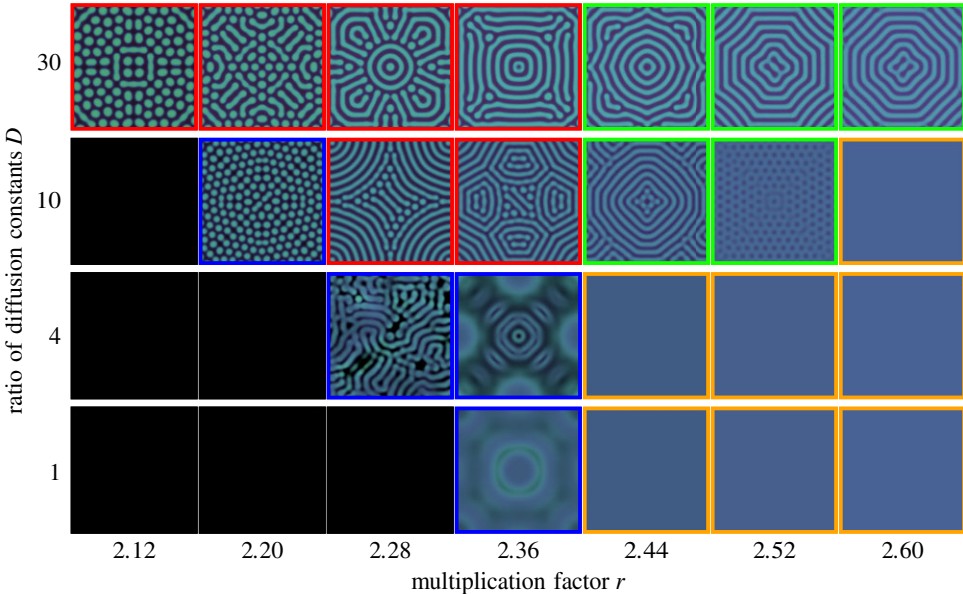

**Figure 1.** Spatial patterns with various parameters (reproduced with permission from [22]). For simulating given reaction–diffusion system, we consider a two-dimensional square lattice. The space is discretized into small sites which we refer to as patches. Each subpopulation resides in each patch where they play the eco-evolutionary public goods game with a maximum group size of $N = 8$. For visualizing, cooperator and defector densities are presented as mint green and fuchsia pink colours and the brightness indicates the total density (see appendix D). There are five phases (framed using different colours), extinction (black), chaos (blue), diffusion-induced coexistence (red), diffusion-induced instability (green) and homogeneous coexistence (orange). Among them, chaos patterns are dynamic while others are stationary patterns. We used the Crank–Nicolson method to get patterns with a linear system size of $L = 283$, $dt = 0.1$ and $dx = 1.4$. All configurations are obtained after at least $t = 10\,000$. A uniform disc with densities $u = v = 0.1$ at a centre is used for an initial condition. We use constant birth rate of $b = 1$ and death rate of $d = 1.2$. Note that the symmetry breaking for $r = 2.28$ and $D = 4$ arises from numerical underflow [33].

dependence. To study the impact of $f$ and $\rho$, we examine all possible combinations of $f$, $1 - f$, $\rho$ and $1 - \rho$ taking into account their geometry. The different cases have different geometries, and thus they cannot span each others. Density-dependent functional forms are visualized in figure 2a in $f$ and $\rho$ space.

Since two distinctly different directions of pattern formation are observed by the density-dependent diffusions, we take a closer look at two representative density-dependent diffusion formulations instead of tracking all functional forms. As possible concrete examples, we develop two relevant formulations; one inspired by bacterial diffusion on a Petri dish and the second inspired by human migration studies [37,47]. These mobility patterns can be described by a subset of functions described in figure 2a. For the diffusion sketched from the bacterial movement, we look at the experiment results and its modelling [36,47]. In the model, bacteria grow by consuming nutrients and spread by diffusing in space. The results have shown that the bacteria grow faster when nutrients are in abundance and slower when the bacterial density is too low. From this experimental result, we interpret that bacterial productivity is fast, when nutrients are abundant, and slow, when bacterial concentrations are too low. Defectors mobility is thus a function of their productivity, approximated as $vwf$ equivalent to $\rho(1 - \rho)f(1 - f)$ from $f = u/(u + v)$ and $\rho = u + v$,

$$D_{d}^{(B)}(f, \rho) = D_{c}[1 + 16\sigma\rho(1 - \rho)f(1 - f)], \tag{2.6}$$

where the factor 16 comes from the normalization of the density-dependent part. As shown in figure 3a, the diffusion coefficient $D_{d}^{(B)}$ has a maximum at intermediate values of $f$ and $\rho$. In contrast to bacteria, human mobility may be maximized at low and high population densities $\rho$. Utility in humans seems to be maximized by avoiding extremely low and extremely high total population densities [37]. We introduce this diffusion dynamics for defectors as,

$$D_{d}^{(H)}(f, \rho) = D_{c}[1 + \sigma\{4\rho(\rho - 1) + 1\}]. \tag{2.7}$$

The addition of 1 maintains non-zero diffusion $D_{d} > 0$.

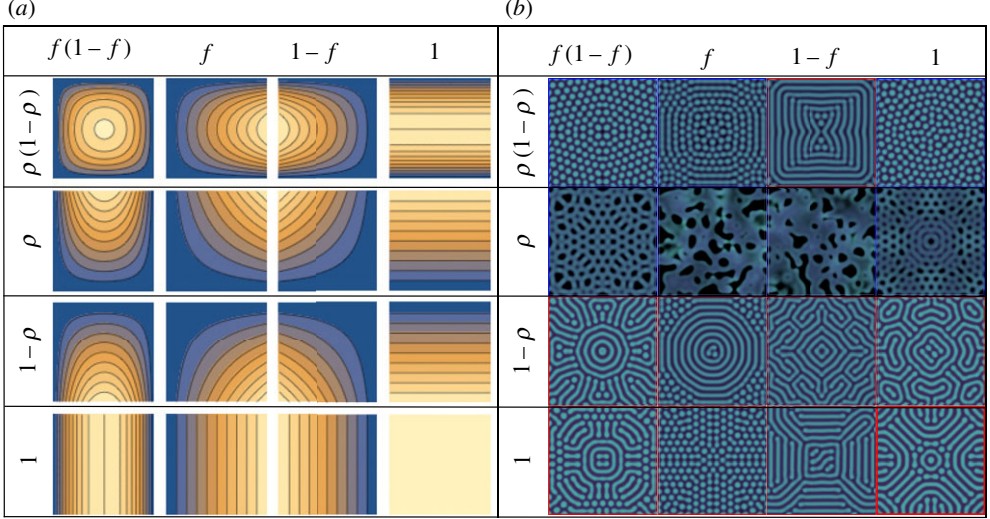

**Figure 2.** Patterns with various functional forms for defector's diffusion coefficient. The density-dependent functional form is determined by multiplying the functions in row and column. In (a), the shape of functions is shown in $f$ ($x$-axis) and $\rho$ ($y$-axis) space (contour plot). Blue and yellow colours represent low and high values at a given $f$ and $\rho$, respectively. In (b), we present the patterns at a given functional form for $r = 2.32$ and $\sigma = 20$. As we can see, different density-dependent diffusion shows different patterns, largely dotted and striped patterns. Here, we use the red and blue coloured frames for striped and dotted patterns, respectively. We include the chaotic patterns in dotted patterns because there chaotic patterns emerge close to the dotted patterns in parameter space (figure 1). A uniform disc with densities $u = v = 0.1$ at a centre is used for an initial condition. Note that symmetry breaking patterns come from numerical underflow [33]. We use the forward Euler method with $dt = 0.005$ and $dx = 1.4$, and the stabilized patterns are obtained after at least $t = 4500$. For the temporal evolution of all these patterns, see electronic supplementary material, video.

To implement the dynamics as in equation (2.4), we numerically solve the equation on a two-dimensional square lattice. The Crank–Nicolson method is used for constant diffusion coefficient; however, we use the forward Euler method for a dynamic diffusion coefficient. Since we are varying $\sigma$, we focus on $r$-$\sigma$ space instead of $r$-$D$ space as opposed to [22]. The diffusion coefficient now can vary in space due to the inhomogeneous densities. An average diffusion coefficient $\bar{D} = \langle D \rangle_{x,y}$ averaged in all patches is determined at a given $\sigma$ and the associated density-dependent dynamics. Diffusion dynamics as described above is a function of both evolutionary (fraction of cooperators) as well as ecological (total population density) parameters. This eco-evolutionary diffusion dependence forms the nucleus of our model elucidating the effects of eco-evolutionary processes on pattern formation.

## 3. Results

We get patterns for $r = 2.32 < r_{\mathrm{hopf}}$ at given 16 density-dependent diffusion coefficient formula and compare the results in figure 2b. Different density dependencies show different patterns, and largely there are two patterns, dot and stripe. As a crude conclusion, with $\rho$ in $D_{\mathrm{d}}(f, \rho)$, dotted or chaotic patterns appear implying defectors' slow movement in low density $\rho$ induces the dotted or chaotic patterns. In general, the dotted and chaotic patterns are observed close to the extinction phase while striped patterns are far from the extinction. Therefore, the emergence of dotted and chaotic patterns implies that the density-dependent diffusion drives the system to the margins of the harsh environment for surviving with $r < r_{\mathrm{hopf}}$.

For more intensive investigation of this different pattern formation, we focus only two relevant and concrete examples formulated in equations (2.6) and (2.7) inspired by bacterial growth and human migration. The bacterial diffusion comes from the top left corner in figure 2a, and it is expected that dotted patterns are favoured in this case. On the other hand, the human migration is not exactly mapping into one of the functional form, but we can expect the results from our analysis above. For humans, defectors move faster when the density $\rho$ is low which is exactly opposite to the behaviour of inducing dotted pattern. Hence, we can infer that the striped patterns are more favoured. As expected, we get the dotted patterns in bacterial diffusion while striped patterns appear in human migration (figure 3).

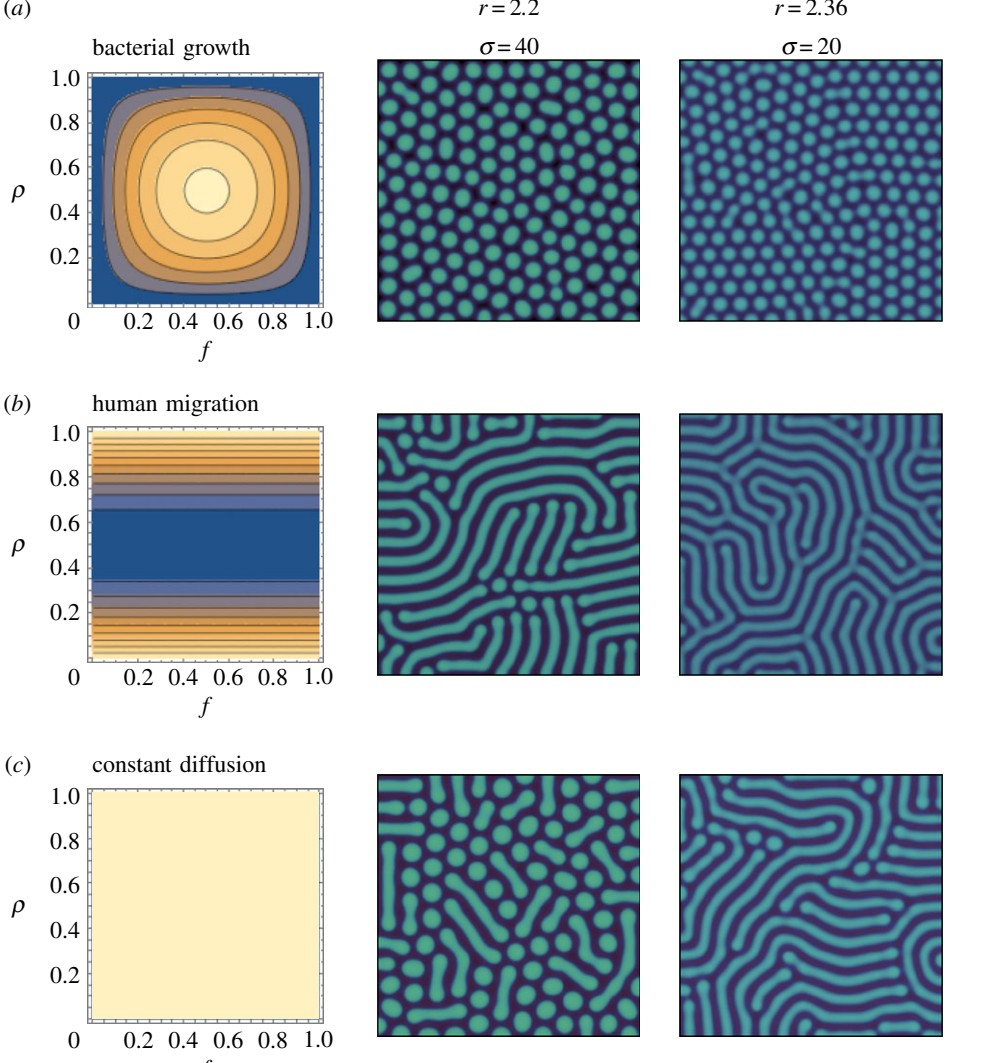

**Figure 3.** Patterns under the two density-dependent behaviours for $r = 2.2$ and $r = 2.36$ (in column) with $\sigma = 40$ and $\sigma = 20$, respectively. The sensitivity to density dependence $\sigma$ is denoted with the multiplication factors $r$. The patterns in the same row as (a) are the result of the density-dependent diffusion function as in equation (2.6). The second row (b) is the results of using equation (2.7). The functions $D_d(f, \rho)$ are shown as contour plots in $f$ and $\rho$ space. Blue and yellow colours represent low and high values at a given $f$ and $\rho$, respectively. For bacterial growth, dotted patterns emerge, while striped patterns appear for human migration. The results show that the fast movement of the defectors in high reproduction region forces the system to the edge of extinction. We use $L = 283$, $N = 8$, $dt = 0.005$, $dx = 1.4$, $b = 1$ and $d = 1.2$. Cooperator and defector densities in each patch are randomly drawn from 0 to 0.1 as an initial condition.

Next, we test the robustness of bacterial and human mobility in the $r$-$\sigma$ space. Bacterial diffusion mainly forms dotted pattern for $r < r_{hopf}$, while human migration shows striped patterns as seen in figure 4. The tendency towards these patterns remains stable even when average diffusion coefficients $\bar{D}$ are the same for both cases. These patterns imply that two different density-dependent diffusions modify the ranges of the surviving area in the $r$-$\sigma$ parameter space. At a given $\sigma$, human migration dynamics is more resilient against extinction than bacterial diffusion. This resilience is lost for $r > r_{hopf}$ because two different density-dependent diffusions show opposite behaviour when $r$ increases (figure 5). For small values of $r$, bacterial diffusion suppresses average diffusion coefficient $\bar{D}$ while human migration diffusion boosts $\bar{D}$. However, the effect is opposite for large $r$ as shown in figure 5. As a result, bacterial diffusion reduces the size of the parameter space where survival is possible—the habitable space—for $r < r_{hopf}$. For $r > r_{hopf}$, bacterial diffusion increases the size of the region with patterns when comparing bacterial and human migration dynamics. Furthermore, we analyse the average quantities of patterns for each case, see appendix C. Interestingly and counterintuitively, for $r < r_{hopf}$, we observe the higher cooperator fraction in the striped patterns than the dotted patterns.

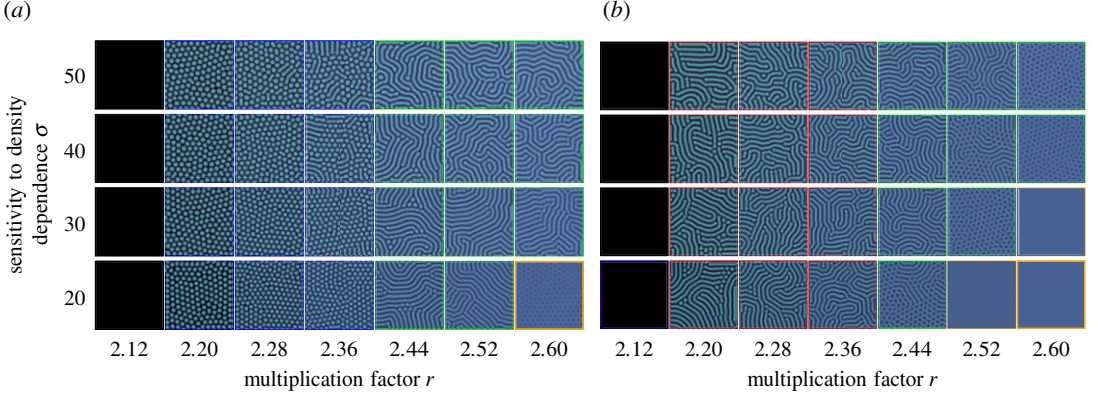

**Figure 4.** Phase diagrams with different diffusion functions: (*a*) where the diffusion reflects bacterial growth equation (2.6) and (*b*) where the diffusion reflects human migration equation (2.7). Each frame of pattern is coloured by the same criteria with figure 1 except dotted pattern for $r < r_{hopf}$. To distinguish dotted and striped patterns for $r < r_{hopf}$, we have used different colours. For dotted patterns, we used the blue colour which is used for chaotic patterns, as both are observed near extinction phase. We can clearly see the different patterns for different density-dependent diffusion for $r < r_{hopf}$. Cooperator and defector densities in each patch are randomly drawn from 0 to 0.1 as an initial condition.

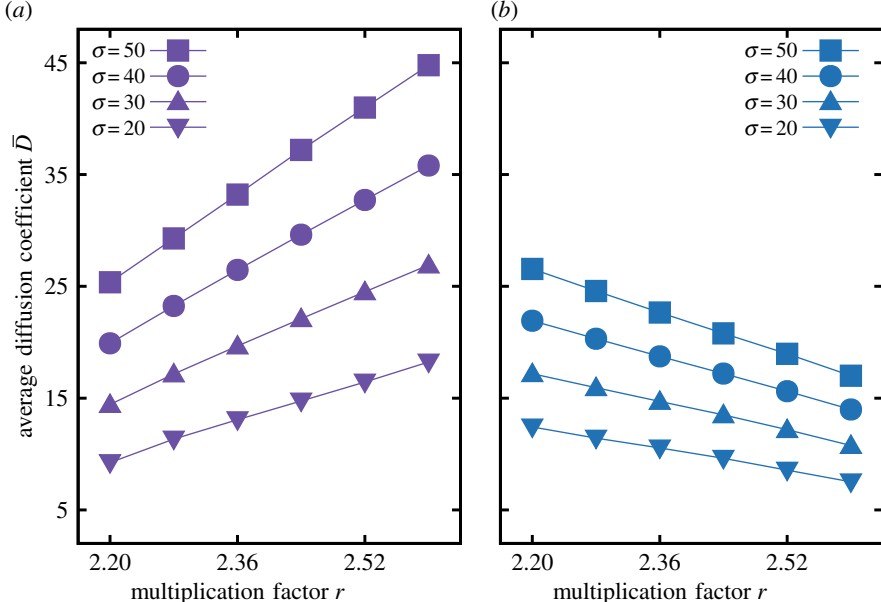

**Figure 5.** (*a*) Bacterial growth, (*b*) human migration. The average diffusion coefficient $\bar{D}$ against the multiplication factor *r* at a given $\sigma$. Different point symbols represent results of $\bar{D}$ with different $\sigma$. Configurations in figure 4 are used for calculating $\bar{D}$. Solid lines are guidelines for points with the same $\sigma$. The average diffusion coefficient $\bar{D}$ increases as *r* increases for bacterial growth diffusion, while $\bar{D}$ decreases for human migration diffusion. This opposite behaviour over *r* of two different diffusions makes different shifts of two boundaries between heterogeneous and homogeneous pattern phases for $r < r_{hopf}$ and $r > r_{hopf}$. For $r < r_{hopf}$, the boundary under bacterial growth diffusion is located much higher than human migration diffusion, while it is opposite for $r > r_{hopf}$.

The different diffusion dynamics for different organisms results in different spatial patterns. Surely, this can result from the diffusion property itself. The defectors grow, reproduce and spread their offspring fast, inducing a fast decrease of cooperators while increasing the risk of extinction leading to a formation of dotted patterns. The tendency of human migration to stay in moderate total density stabilizes populations producing striped patterns implying that diffusion increases the size of the habitable parameter space.

## 4. Summary and discussion

We investigate the effect of diffusion driven by eco-evolutionary dynamics on pattern formation using numerical calculations. We find the kinds of movements that induce dotted or striped patterns

drawing close to extinction phase or rescuing the population from extinction risk. The results show that slow movement of defectors in low total population density threatens the system with extinction. For more intensive analysis, we focus on two opposite examples of density-dependent diffusions, mimicking bacterial growth dynamics and human migration. We confirm our findings by noting the observation that one behaviour draws the system to the edge of survival, whereas the other increases the size of the habitable parameter space. This result supports the hypothesis that a structured population including migration dynamics can help avoid extinctions and facilitate the maintenance of diversity [48,49].

The equations of motion employed in this study are in principle modifications of the classical inhibitor–activator systems [32]. While it is clear that pattern formation is possible due to the higher diffusion coefficient of the inhibitor, we have provided a biologically meaningful reason for this diffusion disparity between activators and inhibitors at a given assumption for the constant diffusion of cooperators. Across scales of organization, it might be possible that defectors, cheaters, cancerous cells etc. have secured higher mobility as a benefit from not paying the costs of cooperation [46,50]. Many of such model systems are *in vivo* based in turbulent environments, e.g. bloodstreams and fluid environments. A control over mobility through density can be envisioned by the evolution of stickiness or such an associated trait which can work against environmental shearing; control of metastasis of cancer cells via cell densities has been recently proposed [51]. Such examples exemplify the higher diffusion coefficient of the defectors over cooperators as we have used in our model. Our model, however, goes a step beyond in including density-dependent diffusion coefficients. A number of studies show why this extension is not only of theoretical interest but could be a widely observed property from classical ecology to sociobiology. Negative density-dependent dispersion (diffusion rate decreases with the total population density) can come about in different species due to a variety of reasons [52]. Avoiding inbreeding depression, competition for resources, resolving sexual conflicts or response to climate change as seen in vole populations [53–55] are just some of the causes. On the human scale, theoretical results of social dilemma resolutions as well as the experimental results should be heeded with caution; cross-cultural studies highlight the difference in social attitudes [56,57]. We believe that conducting social dilemma experiments in cities and their countrysides might already tease out the microstructure in behaviours across the spatial landscape. With this knowledge, it is possible to make educated migration decisions. If a certain location is getting too crowded then it might not be the best option to stay there—a throwback to the classical 'El Farol' problem from Santa Fe [58]. Density-dependent diffusion can influence not just independent levels of the organization but also mediate the transitions between them for e.g. the evolution of multicellularity [59].

Diffusion with a preference for forming alliances can induce ecological conditions which are favourable to the spread of cooperation [60]. We have shown that diffusion driven by eco-evolutionary dynamics is instrumental in generating patterns which can be routinely seen in nature. Density-dependent movement resulting in striking spatial patterns is a well-known phenomena in physics known as the Cahn–Hilliard principle of phase separation. Studies have highlighted its underuse in ecology even though examples satisfying the principle abound in nature from sperm cells to mussel beds [61,62]. The relationship between the movement of mussels and their density is similar to the human migration pattern proposed herein. The resulting patterns are thus similar as well [61], although in our case we have also included population dynamics. Patterns in nature are generally expected to promote efficiency in organisms. Different organisms, with their different eco-evolutionary diffusion dynamics, show different patterns which affect their ability to survive harsh conditions. In this context, despite our strong assumption for the constant diffusion of cooperators, our finding may support the reason why we frequently observe dotted patterns in nature when the population gets stressed, either via extrinsic causes such as the environment or the population composition (e.g. increase in defectors) [63]. On a macro-scale, this further encourages the use of both ecological as well as evolutionary approaches to understanding regular patterns observed widely in nature.

Data accessibility. The core codes, for the Crank–Nicolson algorithm and the forward Euler method, and test gnuplot scripts are available on `Github` at https://github.com/tecoevo/DensityDependentDiffusion.

Authors' contributions. Both authors conceived the project, developed and analysed the model and wrote the paper.

Competing interests. We declare we have no competing interests.

Funding. Both authors acknowledge generous support from the Max Planck Society.

Acknowledgements. We thank Christoph Hauert and three anonymous referees for comments and suggestions for improving the manuscript. We also thank David Rogers for improving the readability and clarity of the manuscript. The authors thank a reviewer from a previous submission for constructive criticism.

# Appendix A. Average payoffs

In a PGG, cooperators invest a fixed amount $c$ in the common pool. The investments of all cooperators are amplified by multiplication factor $r > 1$, and then evenly returned to all individual participants $S$ in the game as the benefit. Under this setting, the payoffs for defectors and cooperators are given by,

$$\left.\begin{aligned} P_D(m; S) &= \frac{rcm}{S} \\ \text{and} \quad P_C(m; S) &= \frac{rcm}{S} - c, \end{aligned}\right\} \tag{A 1}$$

with $m$ cooperators. While we assume that the capacity of group size is $N$, under eco-evolutionary dynamics, it might be impossible that all $N$ individuals participate in the game [26]. Therefore, the actual interacting group size can range from $S = 2$ to $N$. Individuals have a chance to meet and interact with each other with a probability that is proportional to the total density in a well-mixed population. Therefore, the probability $p(S; N)$ that an individual finds itself in a group of size $S$ is given by,

$$p(S; N) = \binom{N-1}{S-1}(1-w)^{S-1}w^{N-S}, \tag{A 2}$$

with sparseness $w = 1 - u - v$. The average payoffs for defectors and cooperators, $f_D$ and $f_C$, are then calculated as follows:

$$\left.\begin{aligned} f_D &= \sum_{S=2}^{N} p(S; N)\overline{P_D}(S) \\ \text{and} \quad f_C &= \sum_{S=2}^{N} p(S; N)\overline{P_C}(S), \end{aligned}\right\} \tag{A 3}$$

where $\overline{P_D}(S)$ and $\overline{P_C}(S)$ are the expected payoffs for defectors and cooperators, respectively, with interacting group size $S$ with the minimum group size being 2.

The probability that there are $m$ cooperators among the $S - 1$ other individuals is given by $p_c(m; S)$,

$$p_c(m; S) = \binom{S-1}{m}\left(\frac{u}{1-w}\right)^m \left(\frac{v}{1-w}\right)^{S-1-m}, \tag{A 4}$$

and the expected payoffs are written as,

$$\left.\begin{aligned} \overline{P_D}(S) &= \sum_{m=0}^{S-1} p_c(m; S)P_D(m; S) = \frac{r}{S}\sum_{m=0}^{S-1} m p_c(m; S) \\ \text{and} \quad \overline{P_C}(S) &= \sum_{m=0}^{S-1} p_c(m; S)P_C(m+1; S) = \frac{r}{S}\sum_{m=0}^{S-1} (m+1)p_c(m; S) - 1, \end{aligned}\right\} \tag{A 5}$$

where we set the investment cost $c = 1$. Accordingly, the average payoffs $f_D$ and $f_C$ are calculated as follows:

$$\left.\begin{aligned} f_D &= \frac{ru}{1-w}\left[1 - \frac{(1-w^N)}{N(1-w)}\right] \\ \text{and} \quad f_C &= f_D - 1 - (r-1)w^{N-1} + \frac{r}{N}\frac{1-w^N}{1-w}. \end{aligned}\right\} \tag{A 6}$$

# Appendix B. Stable fixed point

Without spatial dynamics, the densities of cooperators and defectors change over time as described by the following equations:

$$\dot{u} = \nabla \cdot (D_c \nabla u) + u[w(f_C + b) - d] \tag{B 1a}$$

and

$$\dot{v} = \nabla \cdot (D_d \nabla v) + v[w(f_D + b) - d]. \tag{B 1b}$$

These equations are similar to the classical activator–inhibitor systems [32]. One of the fixed point(s) of this system is stable. Accordingly, the relation between visible $f$ and $\rho$ is imposed. Without spatial dynamics, we can perform a stability analysis for the system when we get an interior stable fixed

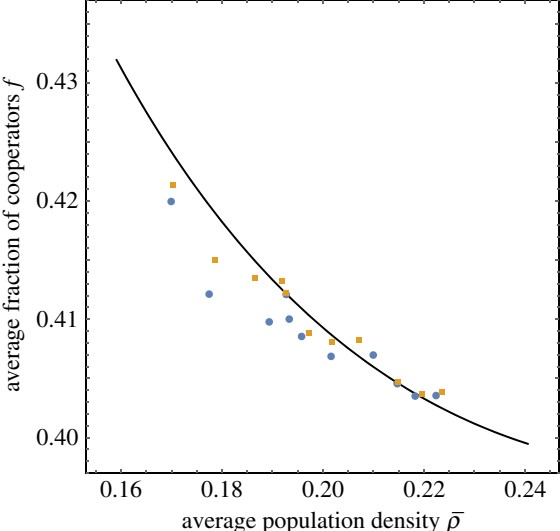

**Figure 6.** Stable fixed points of equation (B 1) for $r > r_{\mathrm{hopf}}$. The cooperator fraction $f$ and total density $\rho$ are in anti-correlation. When one increases, the other decreases. With density-dependent diffusion, the average cooperator fraction $\bar{f}$ and total density $\bar{\rho}$ keeps the relation. The points are the scatter plot of $\bar{f}$ and $\bar{\rho}$ with various $r$. Circles and squares indicate results with bacterial growth and human migration cases, respectively. The solid line is the fixed point evaluated numerically for $N = 8$, $d = 1.2$, and $b = 1$ and $r$ varying from 2.44 to 2.6 with several $\sigma$.

point for $r > r_{\mathrm{hopf}}$. From the solution, we can get the relation between $f$ and $\rho$ at the stable fixed point. As shown in figure 6, $f$ and $\rho$ are in a certain relationship, i.e. the possible $f$ and $\rho$ pairs are restricted. When $\rho$ increases, $f$ decreases along multiplication factor $r$. Now we include spatial dynamics. With density-dependent diffusion, the average $f$ and $\rho$ are in good agreement with the relationship established under no spatial effects for $r > r_{\mathrm{hopf}}$.

## Appendix C. Pattern analysis

Given the predominance of the two types of patterns, dotted and striped, we analyse the different properties between them for $r < r_{\mathrm{hopf}}$. Herein, we used configurations for $\sigma = 18.75$ and $r = 2.36$ with two different density-dependent diffusion functions for analysing patterns. Since $u$ and $v$ are inhomogeneous in $x$-$y$ space, $D_{\mathrm{d}}(u, v)$ is also spatially inhomogeneous.

Counterintuitively, the striped pattern has higher average cooperator fraction $\bar{f}$ than the dotted pattern (0.425 > 0.422). The average total density $\bar{\rho}$ of the striped pattern is smaller than that of the dotted pattern (0.156 < 0.159). It seems that the striped pattern is closer to the extinction phase, because the high fraction of cooperators and small total density are the properties of patterns for small $r$. To understand this result, we examine the spatial distribution of $f$ and $\rho$, because the average of all subpopulations may not be representative.

The two spatial dimensions $x$, $y$ are both of size 283. We pick five $y$ values to look at the spatial distribution of $u$ and $v$, $y = 10$, 71, 141, 211 and 272. These are the values corresponding to regions near the domain boundaries, centre and the intermediate position (figure 7), respectively. We observe that the cooperator density is locally much higher than the defector density at the centre for the dotted pattern. The aforementioned does not appear in striped patterns. Hence, $f$ is larger in the dotted pattern than the striped pattern at the centre. However, there are some places which have smaller $f$ in the dotted pattern than the striped pattern, and thus $\bar{f}$ in the dotted pattern is lower than that of the striped pattern even though they have a higher cooperator fraction locally.

To see this effect, we look at the average cooperator fraction $f_y$ at each slice (average over $x$ at a fixed $y$). As we can see in figure 8, even though $\bar{f}$ for the dotted pattern is smaller than that of the striped pattern, locally $f_y$ has a higher value. The dotted pattern has a much higher deviation of $f$ and thus locally has much higher $f$ than that of the striped pattern even though it has smaller average cooperator fraction $\bar{f}$. It seems that the large fluctuations of $f$ and $\rho$ are the properties of the patterns close to the extinction phase.

For more general analysis of patterns for $r < r_{\mathrm{hopf}}$, we measure the difference between average quantities in different patterns at the same parameter set; we measure the difference $\Delta\bar{f}$ of the average

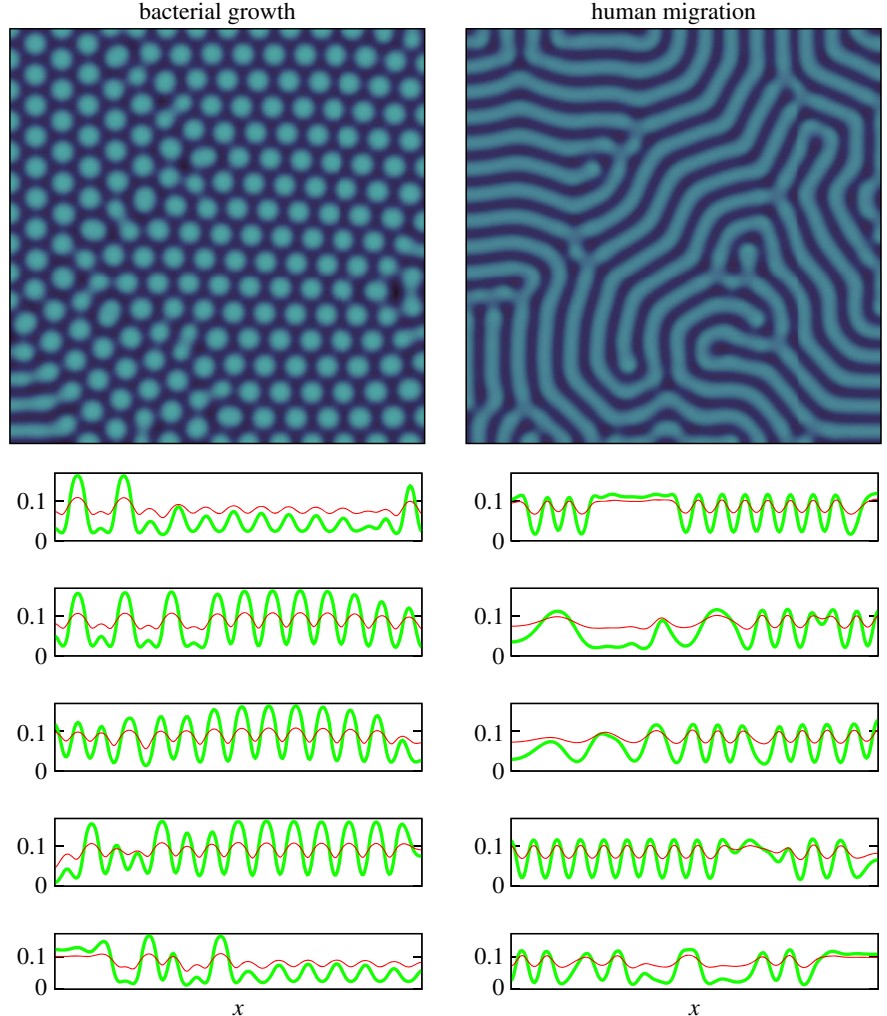

**Figure 7.** Density profiles at a given y. At a given pattern in each column, we horizontally slice along the x-axis at different points such as y = 271, 211, 141, 71 and 10 (from top to bottom) to show defector and cooperator densities. This gives us the densities $u$ and $v$ corresponding to regions near the domain boundaries (272,10), centre (141) and intermediate positions (211,71). The left (right) column is for bacterial growth (human migration) case. Cooperator and defector densities are represented by thick green and thin red coloured lines, respectively.

fractions of cooperators as $\Delta\bar{f} = \bar{f}$ at the dotted pattern $-\bar{f}$ at the striped pattern. In parallel, we also measure the different $\Delta\bar{\rho}$ of the average population density the in the same way. We used all patterns in figure 4. As we can see in figure 9, dotted patterns always get lower cooperator fractions than the striped pattern under the same parameter set while the density can be higher in higher $\sigma$. Interestingly, with decreasing $\sigma$ the total density more rapidly decreases in dotted patterns than striped patterns. This is the evidence that the striped patterns have a larger surviving phase. In figure 9c, we scatter plots all quantities of patterns regardless of parameter sets. Again, we can find the diffusion inspired by bacterial growth give larger $\bar{f}$ at the same $\bar{\rho}$.

## Appendix D. Colour coding

For visualizing cooperator and defector densities, we use mint green (colour code: no. A7FF70) and fuchsia pink (colour code: no. FF8AF3) colours for each type (colour names: [64]). The ratio of cooperators determines the colour spectrum and saturation of the colour. If only cooperators (defectors) are observed, the corresponding site is coloured mint green (fuchsia pink). The Maya blue colour appears when cooperators and defectors have the same density. The total density of the population is represented as the brightness of the colour. For convenience, we use HSB colour space which is a cylindrical coordinate system $(r, \theta, h) =$ (saturation, hue, brightness). The radius of circle $r$

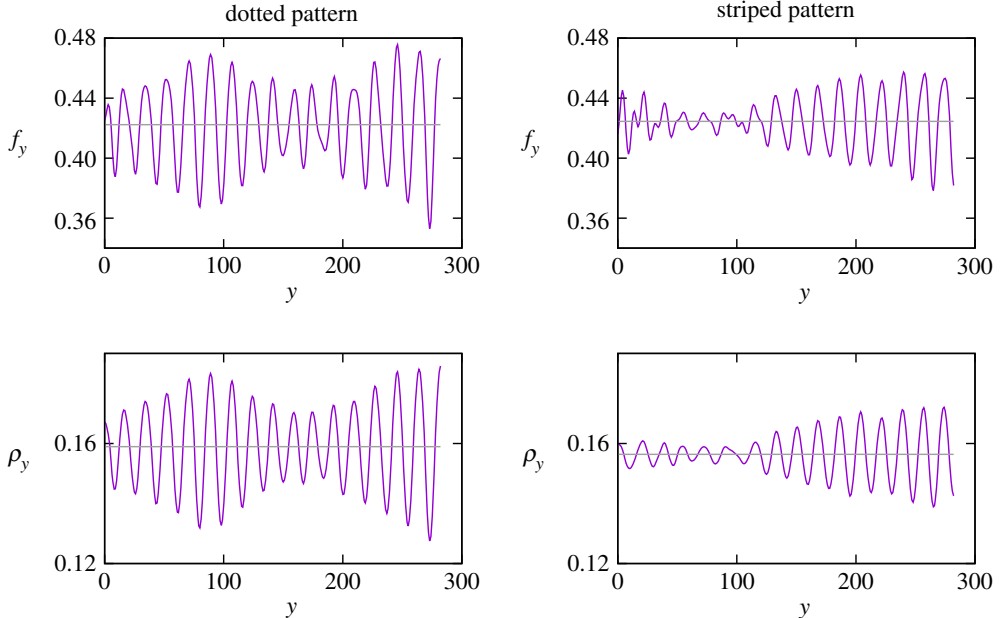

**Figure 8.** General $y$. The left column is for the dotted pattern, while right one is for the striped pattern. We can see that $f_y$ and $\rho_y$, are fluctuating in $y$, and the dotted pattern has a much larger deviation than the striped pattern. The dotted patterns locally have higher $f_y$ than striped patterns.

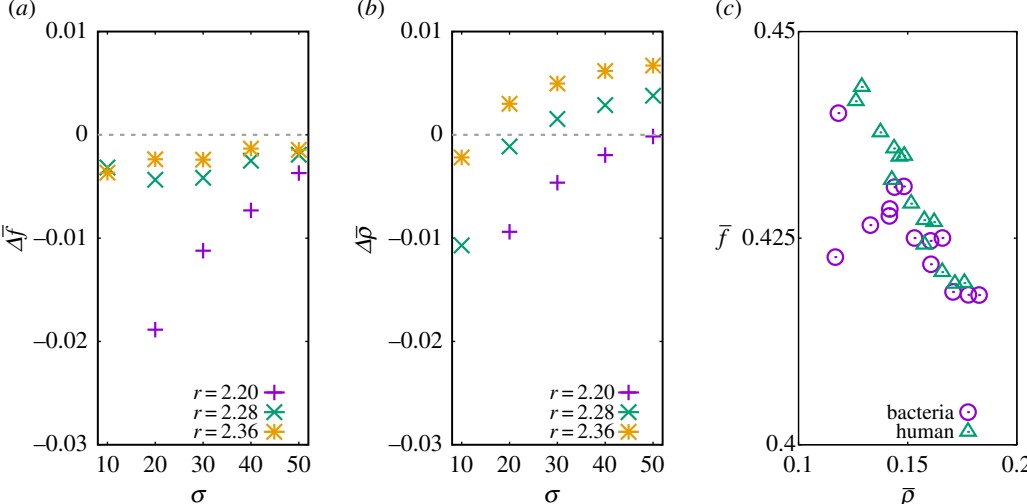

**Figure 9.** The difference of average quantities in dotted and striped patterns. For patterns used in figure 4, we try to figure out the distinct properties of dotted and striped patterns. The differences of average quantities $\Delta\bar{f}$ and $\Delta\bar{\rho}$ get from the average quantity of dotted and striped patterns. The negative $\Delta\bar{f}(\Delta\bar{\rho})$ indicates the higher $\bar{f}(\bar{\rho})$ of the striped patterns than the dotted patterns.

indicates saturation or the colour. The angular variable $\theta$ represents the colour spectrum, hue where the red, green and blue are located at $\theta = 0$, $\theta = 2\pi/3$ and $\theta = 4\pi/3$, respectively. The relation between RGB and HSB coordinate is $\tan(\theta) = (\sqrt{3} \cdot (G - B))/(2 \cdot R - G - B)$, where $R$, $G$ and $B$ are red, green and blue coordinates of RGB space (cartesian coordinate). By blending two different colours which have different $r$ and $\theta$, we denote the cooperator fraction.

The height $h$ indicates the brightness of the colour. Here, we use brightness as for the total population density $\rho = u + v$. If we use a linear function of $\rho$ for brightness, it is hard to figure out the patterns for small population densities. For better visualization, we formulate the brightness as

$$\frac{\log a\rho + 1}{\log a + 1}, \tag{D1}$$

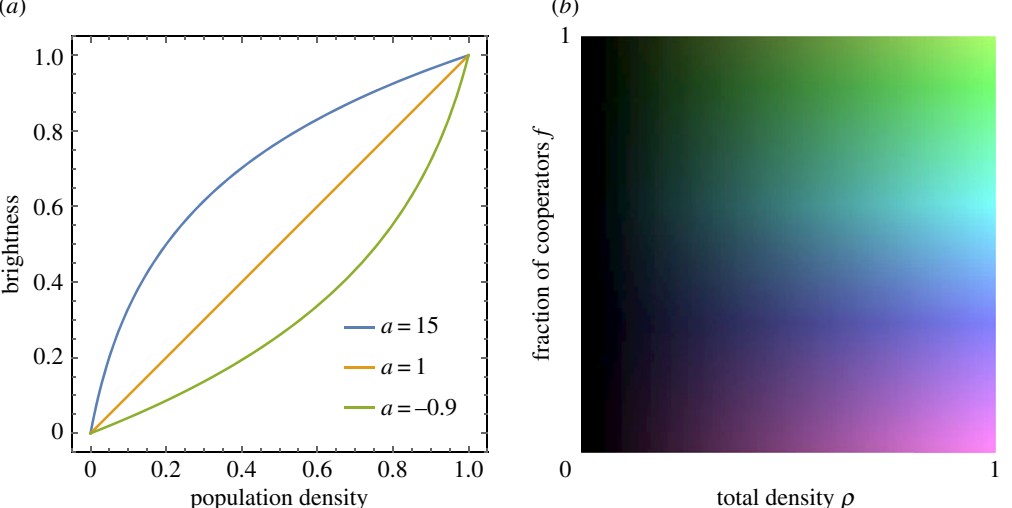

**Figure 10.** (*a*) To amplify brightness for small population densities, we use a nonlinear relation between population density $\rho$ and brightness. When $a = 1$, the linear relation is recovered. For all figures in the main script, we use $a = 15$. (*b*) The exact colour scheme used for colouring the patterns. Each position in a pattern is coloured using this palette by choosing the corresponding $f$ and $\rho$ values.

where a control parameter $a$ ($> -1$ and $\neq 0$) determines the curvature of the brightness function in the total population density $\rho$ (figure 10). The complete colour scheme withstands the standard tests for colour blindness.

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
