## [Reviewer comments · Royal Society Open Science]

Review History

RSOS-181273.R0 (Original submission)

Review form: Reviewer 1

Is the manuscript scientifically sound in its present form?

Yes

Are the interpretations and conclusions justified by the results?

No

Is the language acceptable?

Yes

Is it clear how to access all supporting data?

Not Applicable

Do you have any ethical concerns with this paper?

No

Have you any concerns about statistical analyses in this paper?

No

Recommendation?

Major revision is needed (please make suggestions in comments)

Comments to the Author(s)

This paper studies the effect of non-constant diffusion rate in pattern formation in spatial ecological public goods games. The topic and their results are very interesting, but I have some major concerns.

Major 1

For bacterial diffusion model, the authors assume that cooperator diffusion rate is constant and that defector diffusion rate increases as f (fraction of cooperator) and ρ (total density) approach $1/2$. They provide some explanations why this mimics bacterial diffusion, which I could not understand at all.

Generally speaking, there are many species of bacteria and calling it bacterial diffusion model might be too much. Different bacteria species produce different patterns.

Major 2

Cooperator diffusion rate is set constant throughout the paper. This is probably much stronger assumption than the authors supposed. If cooperators in different positions have different diffusion rates, we cannot rescale them as unity without loss of generality. So the family of systems studied in this paper contains very special cases only.

Since the system is very complicated, I do not insist to redo the whole analysis. However, some sentences, particularly in Section 4, are claiming far more than the present analysis can suggest. Texts should be modified or weakened accordingly.

Major 3

Figure 2 must be comparable with Figure 1. I have once worked with this system, so I know that numerical treatment of this system is quite sensitive. Patterns could be different in Crank-Nicholson and Euler methods. Thus, the comparison with the baseline case is important. See my comments below.

Other Comments:

All figures

It is better to mention which snapshot pattern is stationary and which one is dynamic.

Fig.2

I do not see why $r=2.32$ is used. This parameter is not shown in Fig 1 (baseline constant diffusion model), so readers cannot directly compare. It is very important to check if the authors' implementation of non-constant diffusion is working good or not.

I also do not understand the orange box at right-bottom in Fig.2b. Is this coexistence? Extinction? Coexistence in $r=2.32$ with constant diffusion rate is quite unlikely.

above (2.7)

$v w f = \rho (1-\rho) f (1-f)$ from (2.2)

I do not understand this part. From where do we get this equation?
Do you mean $v w f =$ (defector productivity) ?

Section 4:

"Slow movement in" -> "Slow movement OF DEFECTORS in ..."

Acknowledgements, line 3

The authors thank a reviewer ...

Review form: Reviewer 2 (Christoph Hauert)

Is the manuscript scientifically sound in its present form?

Yes

Are the interpretations and conclusions justified by the results?

Yes

Is the language acceptable?

Yes

Is it clear how to access all supporting data?

Yes

Do you have any ethical concerns with this paper?

No

Have you any concerns about statistical analyses in this paper?

No

Recommendation?

Accept with minor revision (please list in comments)

Comments to the Author(s)

Summary:

The authors present an interesting and relevant extension to the dynamics of spatial ecological public goods games. Their work is a direct extension of earlier work by Wakano et al PNAS (2009) to non-constant diffusion. In a very thorough manner the authors investigate the pattern formation process for a number of functional forms for the dependence of diffusion on population density or frequency of cooperators. Particular emphasis is placed on two biologically motivated scenarios relating to bacterial and human migration, which result in distinctly different spatial patterns. Overall the study makes an interesting contribution with a number of smaller points that would help to improve the accessibility of the manuscript.

Minor points:

- p3,l56: the carrying capacity is not 1! Instead, 1 is the maximum density, which represents a hard non-sustainable upper limit.
- p4,l12: explain/ motivate lower bound of 2 for S. The 'group' size can be $S=1$.
- p4,l21: identifying w as 'space' is misleading given that explicit spatial dimensions are introduced shortly afterwards. Maybe use 'reproductive opportunities'?

- p4,l46f: it might be worth pointing out that the PDE is actually a continuum limit of a meta-population with sub-populations arranged in a lattice and connected by migration (diffusion) but not through interactions.
- p6,l6-49: the analysis of different functional forms of the diffusion is interesting but a bit exhaustive. Maybe relegate to an appendix?
- p6,l57: it would be helpful if the approximation was briefly explained.
- eq.2.7/8: the $N[\dots]$ is not really needed and can be easily evaluated to make the equations more easily accessible.
- p7,l8: what is the evidence for humans to avoid high densities, i.e. high mobility? The process of urbanization is clearly happening but this suggests a complementary 'ruralization'.
- p8,l8f: see note above (p6,l6-49)
- p8,l19: explain meaning of 'surface area' as there is no boundary between cooperator and defector domains.
- p8,l21: The sentence 'The striped patterns due to (1-f) boost up cooperation from low fractions.' is hard to understand because diffusion of cooperators is constant and the origin of the 1-f term remains unclear.
- fig4: much more illustrative than fig3 because it demonstrates the robustness of patterns.
- p8,l46f: what about the average population density? Is it indeed higher in striped than in dotted patterns as this suggests? This is a key question and fig.C2 points in that direction but certainly deserves to be covered in the main text.
- p8: the first paragraph of the discussion is a bit repetitive.
- p10,l10f: this paragraph sketches interesting potential links but remains rather vague and speculative. Less examples but more specific ties to the presented model and results would be preferable.

Decision letter (RSOS-181273.R0)

18-Sep-2018

Dear Dr Gokhale,

The editors assigned to your paper ("Ecological feedback on diffusion dynamics") have now received comments from reviewers. We would like you to revise your paper in accordance with the referee and Associate Editor suggestions which can be found below (not including confidential reports to the Editor). Please note this decision does not guarantee eventual acceptance.

Please submit a copy of your revised paper before 11-Oct-2018. Please note that the revision deadline will expire at 00.00am on this date. If we do not hear from you within this time then it will be assumed that the paper has been withdrawn. In exceptional circumstances, extensions may be possible if agreed with the Editorial Office in advance. We do not allow multiple rounds of revision so we urge you to make every effort to fully address all of the comments at this stage. If deemed necessary by the Editors, your manuscript will be sent back to one or more of the original reviewers for assessment. If the original reviewers are not available, we may invite new reviewers.

- Data accessibility

If you wish to submit your supporting data or code to Dryad (<http://datadryad.org/>), or modify your current submission to dryad, please use the following link:
<http://datadryad.org/submit?journalID=RSOS&manu=RSOS-181273>

- Competing interests

- Authors' contributions

- Acknowledgements

- Funding statement

Please note that Royal Society Open Science charge article processing charges for all new submissions that are accepted for publication. Charges will also apply to papers transferred to Royal Society Open Science from other Royal Society Publishing journals, as well as papers submitted as part of our collaboration with the Royal Society of Chemistry (<http://rsos.royalsocietypublishing.org/chemistry>). If your manuscript is newly submitted and subsequently accepted for publication, you will be asked to pay the article processing charge, unless you request a waiver and this is approved by Royal Society Publishing. You can find out more about the charges at <http://rsos.royalsocietypublishing.org/page/charges>. Should you have any queries, please contact openscience@royalsociety.org.

on behalf of Dr Andrew Angel (Associate Editor) and Prof. Kevin Padian (Subject Editor)
openscience@royalsociety.org

Associate Editor's comments (Dr Andrew Angel):

Associate Editor: 1

Comments to the Author:

Both of the reviewers were positive about the overall scientific soundness of the manuscript and its importance to the field. However, one of the reviewers highlighted concerns that some of the interpretations and conclusions require some additional work and clarification which go beyond minor corrections. Therefore, I am recommending this manuscript undergo major revision to satisfy the concerns of the reviewer.

Associate Editor: 2

Comments to the Author:

I am recommending the manuscript for peer review.

Comments to Author:

Reviewers' Comments to Author:

Reviewer: 1

Comments to the Author(s)

This paper studies the effect of non-constant diffusion rate in pattern formation in spatial ecological public goods games. The topic and their results are very interesting, but I have some major concerns.

Major 1

For bacterial diffusion model, the authors assume that cooperator diffusion rate is constant and that defector diffusion rate increases as f (fraction of cooperator) and ρ (total density) approach $1/2$. They provide some explanations why this mimics bacterial diffusion, which I could not understand at all.

Generally speaking, there are many species of bacteria and calling it bacterial diffusion model might be too much. Different bacteria species produce different patterns.

Major 2

Cooperator diffusion rate is set constant throughout the paper. This is probably much stronger assumption than the authors supposed. If cooperators in different positions have different diffusion rates, we cannot rescale them as unity without loss of generality. So the family of systems studied in this paper contains very special cases only.

Since the system is very complicated, I do not insist to redo the whole analysis. However, some sentences, particularly in Section 4, are claiming far more than the present analysis can suggest. Texts should be modified or weakened accordingly.

Major 3

Figure 2 must be comparable with Figure 1. I have once worked with this system, so I know that numerical treatment of this system is quite sensitive. Patterns could be different in Crank-Nicholson and Euler methods. Thus, the comparison with the baseline case is important. See my comments below.

Other Comments:

All figures

It is better to mention which snapshot pattern is stationary and which one is dynamic.

Fig.2

I do not see why $r=2.32$ is used. This parameter is not shown in Fig 1 (baseline constant diffusion model), so readers cannot directly compare. It is very important to check if the authors' implementation of non-constant diffusion is working good or not.

I also do not understand the orange box at right-bottom in Fig.2b. Is this coexistence? Extinction? Coexistence in $r=2.32$ with constant diffusion rate is quite unlikely.

above (2.7)

$$v w f = \rho (1-\rho) f (1-f) \text{ from (2.2)}$$

I do not understand this part. From where do we get this equation?

Do you mean $v w f =$ (defector productivity) ?

Section 4:

"Slow movement in" -> "Slow movement OF DEFECTORS in ..."

Acknowledgements, line 3

The authors thank a reviewer ...

Reviewer: 2

Comments to the Author(s)

Summary:

The authors present an interesting and relevant extension to the dynamics of spatial ecological public goods games. Their work is a direct extension of earlier work by Wakano et al PNAS (2009) to non-constant diffusion. In a very thorough manner the authors investigate the pattern formation process for a number of functional forms for the dependence of diffusion on population density or frequency of cooperators. Particular emphasis is placed on two biologically motivated scenarios relating to bacterial and human migration, which result in distinctly different spatial patterns. Overall the study makes an interesting contribution with a number of smaller points that would help to improve the accessibility of the manuscript.

Minor points:

- p3,l56: the carrying capacity is not 1! Instead, 1 is the maximum density, which represents a hard non-sustainable upper limit.
- p4,l12: explain/motivate lower bound of 2 for S. The 'group' size can be $S=1$.
- p4,l21: identifying w as 'space' is misleading given that explicit spatial dimensions are introduced shortly afterwards. Maybe use 'reproductive opportunities'?
- p4,l46f: it might be worth pointing out that the PDE is actually a continuum limit of a meta-population with sub-populations arranged in a lattice and connected by migration (diffusion) but not through interactions.
- p6,l6-49: the analysis of different functional forms of the diffusion is interesting but a bit exhaustive. Maybe relegate to an appendix?
- p6,l57: it would be helpful if the approximation was briefly explained.
- eq.2.7/8: the $N[\dots]$ is not really needed and can be easily evaluated to make the equations more easily accessible.
- p7,l8: what is the evidence for humans to avoid high densities, i.e. high mobility? The process of urbanization is clearly happening but this suggests a complementary 'ruralization'.
- p8,l8f: see note above (p6,l6-49)
- p8,l19: explain meaning of 'surface area' as there is no boundary between cooperator and defector domains.
- p8,l21: The sentence 'The striped patterns due to $(1-f)$ boost up cooperation from low fractions.' is hard to understand because diffusion of cooperators is constant and the origin of the $1-f$ term remains unclear.
- fig4: much more illustrative than fig3 because it demonstrates the robustness of patterns.
- p8,l46f: what about the average population density? Is it indeed higher in striped than in dotted patterns as this suggests? This is a key question and fig.C2 points in that direction but certainly deserves to be covered in the main text.
- p8: the first paragraph of the discussion is a bit repetitive.

- p10,110f: this paragraph sketches interesting potential links but remains rather vague and speculative. Less examples but more specific ties to the presented model and results would be preferable.

Author's Response to Decision Letter for (RSOS-181273.R0)

See Appendix A.

RSOS-181273.R1 (Revision)

Review form: Reviewer 1

Is the manuscript scientifically sound in its present form?

Yes

Are the interpretations and conclusions justified by the results?

Yes

Is the language acceptable?

Yes

Is it clear how to access all supporting data?

Yes

Do you have any ethical concerns with this paper?

No

Have you any concerns about statistical analyses in this paper?

No

Recommendation?

Accept as is

Comments to the Author(s)

The authors have substantially revised the manuscript.
All of my concerns are adequately answered.
I like this paper and I recommend the publication as is.

Decision letter (RSOS-181273.R1)

09-Jan-2019

Dear Dr Gokhale,

I am pleased to inform you that your manuscript entitled "Ecological feedback on diffusion dynamics" is now accepted for publication in Royal Society Open Science.

on behalf of Dr Andrew Angel (Associate Editor) and Professor Kevin Padian (Subject Editor)
openscience@royalsociety.org

Associate Editor Comments to Author (Dr Andrew Angel):

Associate Editor: 1

Comments to the Author:

The reviewer is satisfied that the corrections have addressed their original concerns. Therefore, I am recommending that the manuscript now be accepted as is.

Associate Editor: 2

Comments to the Author:

Thank you for responding to the referees' comments. As major revisions were requested, I am recommending that the manuscript go back to review to ensure that the changes have satisfied those concerns.

Reviewer comments to Author:

Reviewer: 1

Comments to the Author(s)

The authors have substantially revised the manuscript.

All of my concerns are adequately answered.

I like this paper and I recommend the publication as is.

Appendix A

We thank the Editor for providing us with an extension on the resubmission date as this provided us with enough time to run the extra simulations by which we could confidently answer the queries of the reviewers.

=====
Reviewer: 1
=====

Comments to the Author(s)

This paper studies the effect of non-constant diffusion rate in pattern formation in spatial ecological public goods games. The topic and their results are very interesting, but I have some major concerns.

We thank the referee for the thorough review of our manuscript and constructive comments. In our revision we have addressed all the below concerns, we hope in a satisfactory manner.

Major 1

For bacterial diffusion model, the authors assume that cooperator diffusion rate is constant and that defector diffusion rate increases as f (fraction of cooperator) and ρ (total density) approach $1/2$. They provide some explanations why this mimics bacterial diffusion, which I could not understand at all.

Generally speaking, there are many species of bacteria and calling it bacterial diffusion model might be too much. Different bacteria species produce different patterns.

- As we introduced in Eq. (2.5), the density dependent function $g(f, \rho)$ can take any form. Since there are many possibilities for formulating the function, we intend to formulate two different density dependencies that may be observed in reality as concrete examples for showing different pattern formations. We do not intend to insist that our formulation for bacterial diffusion explain general bacterial diffusion. We have formulated the current model according to one particular experimental paper and the diffusion function derived from therein.

As per your comment, we toned down our general claim and explicitly mention that the bacterial diffusion is one possible example inspired by the conventional bacterial growth model and related experiments below Eq. (2.5): "As possible concrete examples, we develop two relevant formulations; one inspired by bacterial diffusion on a petri dish and the second inspired by human migration studies" We now also add more explanation for the new Eq. (2.6): "These mobility patterns can be described by a subset of functions described in Fig 2 (a). For the diffusion sketched from the bacterial movement, we look at the experiment results and its

modeling [36,47]. In the model, bacteria grow by consuming nutrients and spread by diffusing in space.

The results have shown that the bacteria grow faster when nutrients are in abundance and slower when the bacterial density is too low.

From this experimental result, we interpret that bacterial productivity is --- fast, when nutrients are abundant and slow, when bacterial concentrations are too low. ”

Major 2

Cooperator diffusion rate is set constant throughout the paper. This is probably much stronger assumption than the authors supposed. If cooperators in different positions have different diffusion rates, we cannot rescale them as unity without loss of generality. So the family of systems studied in this paper contains very special cases only.

Since the system is very complicated, I do not insist to redo the whole analysis. However, some sentences, particularly in Section 4, are claiming far more than the present analysis can suggest. Texts should be modified or weakened accordingly.

Taking your suggestion into account, we explicitly mention our strong assumption and, accordingly, tone down our claims towards generality: “While it is clear that pattern formation is possible due to the higher diffusion coefficient of the inhibitor, we have provided a biologically meaningful reason for this diffusion disparity between activators and inhibitors at a given assumption for the constant diffusion of cooperators.”, “In this context, despite our strong assumption for the constant diffusion of cooperators, our finding may support the reason why we frequently observe dotted patterns in nature when the population gets stressed, either via extrinsic causes such as the environment or the population composition (e.g. increase in defectors) [63].”

Major 3

Figure 2 must be comparable with Figure 1. I have once worked with this system, so I know that numerical treatment of this system is quite sensitive. Patterns could be different in Crank-Nicholson and Euler methods. Thus, the comparison with the baseline case is important. See my comments below.

We thank the reviewer for this comment. We have reproduced the fig. 1 for showing the overall dynamics in the spatial public goods game, which kind of patterns are obtained in each region. In the main text, on the other hand, we have focused more on the comparison of two density dependent diffusions and have excluded the constant diffusion coefficient. However, as

you pointed out, it would be important to compare the density dependent results with the constant diffusion results, as well. Accordingly, we added a new panel in each figure to compare the results (we also used the forward Euler method to get the results of the constant diffusion to prevent the difference arise from using different methods itself.). For the robustness of results for using two different algorithms, the Crank-Nicolson and the forward Euler methods, we get the same trend in stationary patterns agreeing with the work by Wakano et al PNAS (2009).

Other Comments:

All figures

It is better to mention which snapshot pattern is stationary and which one is dynamic.

We added an explanation “chaos patterns are dynamic while others are stationary patterns.” in the caption of Fig. 1.

Fig.2

I do not see why $r=2.32$ is used. This parameter is not shown in Fig 1 (baseline constant diffusion model), so readers cannot directly compare. It is very important to check if the authors' implementation of non-constant diffusion is working good or not.

We have used $r=2.32$ to clearly see the different pattern formations for different density dependencies. For comparing the baseline constant diffusion model, we added a new panel for the results from the constant diffusion.

I also do not understand the orange box at right-bottom in Fig.2b. Is this coexistence? Extinction?

Coexistence in $r=2.32$ with constant diffusion rate is quite unlikely.

We have not inserted the figure for the constant diffusion result in Fig. 2b to focus on comparisons between density dependent functions only. Again, taking your suggestion into account, we also added a panel for the constant diffusion, and it is coexistence. In Fig1., even for $r=2.28$ and $D=10$, the striped pattern emerges, and the coexistence for $r=2.32$ and $D=20$ makes sense.

above (2.7) $v w f = \rho (1-\rho) f (1-f)$ from (2.2) I do not understand this part. From where do we get this equation?

Do you mean $v w f =$ (defector productivity) ?

Approximately defector productivity can be written as $v w f$, and it can be rewritten by $\rho(1 - \rho)f(1 - f)$ from the definition of f and ρ . We more explicitly wrote the sentence: "Defectors mobility is thus a function of their productivity, approximated as $v w f$ equivalent to $\rho(1 - \rho)f(1 - f)$ from $f = u$ and $\rho = u + v, \dots$ ". Also as per the second referee's comment we now explain the approximation itself.

Section 4:

"Slow movement in" -> "Slow movement OF DEFECTORS in ..."

Thank you. We have made the change.

Acknowledgements, line 3

The authors thank a reviewer ...

Thank you for spotting the typo. Corrected.

=====
Reviewer: 2
=====

Comments to the Author(s)

Summary:

The authors present an interesting and relevant extension to the dynamics of spatial ecological public goods games. Their work is a direct extension of earlier work by Wakano et al PNAS (2009) to non-constant diffusion. In a very thorough manner the authors investigate the pattern formation process for a number of functional forms for the dependence of diffusion on population density or frequency of cooperators. Particular emphasis is placed on two biologically motivated scenarios relating to bacterial and human migration, which result in distinctly different spatial patterns. Overall the study makes an interesting contribution with a number of smaller points that would help to improve the accessibility of the manuscript.

We thank the referee for providing a thorough review of our manuscript and for constructive comments. The suggestions have helped us improve the manuscript immensely.

Minor points:

p3,l56: the carrying capacity is not 1! Instead, 1 is the maximum density, which represents a hard non-sustainable upper limit.

As you correctly pointed out, it is a maximum density. Accordingly, we have changed the statement. Instead of "carrying capacity" we use "maximum density".

p4,l12: explain/motivate lower bound of 2 for S. The 'group' size can be $S=1$.

We provide an explanation for the lower bound 2 above Eq (1): "The lower bound 2 is natural because we need at least two individuals to interact. If there is only one individual, there is no interaction, the game is not played."

p4,l21: identifying w as 'space' is misleading given that explicit spatial dimensions are introduced shortly afterwards. Maybe use 'reproductive opportunities'?

Thank you. Now we explicitly mention "reproductive opportunity" instead of "space" to avoid misleading the readers.

p4,l46f: it might be worth pointing out that the PDE is actually a continuum limit of a meta-population with sub-populations arranged in a lattice and connected by migration (diffusion) but not through interactions.

We are grateful for pointing this out. Above Eq. (2.4), we restate and add an explanation for this: “To include spatial dynamics, we envision subpopulations spatially arranged on a two-dimensional lattice. In each patch, the dynamics of the subpopulation is described by Eq (2.2), and individuals, cooperators and defectors, randomly move between adjacent patches. There is no game interaction between individuals who live in different patches. By taking the continuum limit of this spatially structured subpopulations, we can get the changes of densities over time”

p6,l6-49: the analysis of different functional forms of the diffusion is interesting but a bit exhaustive. Maybe relegate to an appendix?

We thank the reviewer for this suggestion and we fully agree with you. At the same time, we would like to take an approach from the general formula to specific cases. As a compromise we have now massively simplified the explanation for the general case and focus on the two concrete examples. The motivation is captured below Eq. (2.5) as: “Since two distinctly different directions of pattern formation are observed by the density dependent diffusions, we take a closer look at two representative density dependent diffusion formulations instead of tracking all functional forms. As concrete examples, we develop two relevant formulations which can be inspired by real bacterial diffusion on a petri dish and human migration”

p6,l57: it would be helpful if the approximation was briefly explained.

We carefully guess you meant the page 5. We added more explanation for new Eq. (2.6): "These mobility patterns can be described by a subset of functions described in Fig 2 (a). For the diffusion sketched from the bacterial movement, we look at the experiment results and its modeling [36,47]. In the model, bacteria grow by consuming nutrients and spread by diffusing in space. The results have shown that the bacteria grow faster when nutrients are in abundance and slower when the bacterial density is too low. From this experimental result, we interpret that bacterial productivity is --- fast, when nutrients are abundant and slow, when bacterial concentrations are too low.”

eq.2.7/8: the $N[\dots]$ is not really needed and can be easily evaluated to make the equations more easily accessible.

We directly wrote the equations without $N[\dots]$ notation. Accordingly, new Eqs. (2.6) and (2.7) have been edited.

p7,l8: what is the evidence for humans to avoid high densities, i.e. high mobility? The process of urbanization is clearly happening but this suggests a complementary 'ruralization'.

Thanks a lot for pointing this fact out. The reference [37] has shown that even humans prefer to stay longer in the moderate density, thus strong spatial segregation—representing urbanization and ruralization at the same time—can be induced when each individual pursues their own utility. Thus indeed, our suggested density dependent diffusion, high mobility in high density, can explain the human mobility including ruralization. We now include a sentence towards this hypothesis.

p8,l8f: see note above (p6,l6-49)

We believe the comment refers to Fig. 2. As mentioned above, we approach from the general scheme to specific cases. Hence, we would like to keep the figure but simplify the statements and focus on two different density dependent diffusions.

p8,l19: explain meaning of 'surface area' as there is no boundary between cooperator and defector domains.

Indeed. We meant the domain between subpopulations. So far, we have cut down on this part to avoid exhaustive explanation for the general diffusion formulations part.

p8,l21: The sentence 'The striped patterns due to $(1-f)$ boost up cooperation from low fractions.' is hard to understand because diffusion of cooperators is constant and the origin of the $1-f$ term remains unclear.

If the populations have low f (fraction of cooperators), defectors move faster when it contains $(1-f)$ term in the formula of $g(f, \rho)$. Since the population with a faster movement of defectors is rescued forming the pattern, we had made the above statement. As a part of reducing the explanation on general diffusion functions, this statement has also been removed.

fig4: much more illustrative than fig3 because it demonstrates the robustness of patterns.

Thank you.

p8,l46f: what about the average population density? Is it indeed higher in striped than in dotted patterns as this suggests? This is a key question and fig.C2 points in that direction but certainly deserves to be covered in the main text.

We agree with your comment. We measure several quantities of patterns to figure out the underlying mechanisms. While the dotted and striped patterns do show distinct properties, the exact mechanisms underlying their generation are hard to decipher. Thus, in the main text, we emphasize the different pattern formations under diffusion behaviour while relegating the detailed analysis of patterns to the Appendix. As a compromise we have mentioned it more clearly in the main text: “Furthermore, we analyze the average quantities of patterns for each case, see Appendix 4. Interestingly and counterintuitively, for ρ , we observe the higher cooperator fraction in the striped patterns than the dotted patterns.

p8: the first paragraph of the discussion is a bit repetitive.

Thank you. In rephrasing the discussion we have removed the redundancy.

p10,l10f: this paragraph sketches interesting potential links but remains rather vague and speculative. Less examples but more specific ties to the presented model and results would be preferable.

We aim to make use of this paragraph to justify why we had chosen to pursue the line of thought—the extension of diffusion dynamics. Firstly, we highlight why it makes sense that defectors have a higher diffusion coefficient than cooperators. This needs to be highlighted as an essential requirement for pattern formation; the inhibitors show higher diffusion than the activators. Secondly, we highlight the reason why it is crucial to consider density-dependent diffusion coefficients. The way we wrote the paragraph, it might have seemed more like a collection of examples. Taking into account this comment, we now explicitly mention the reason why we focus on the examples (sentences after reference [51]).

We hope the changes we have made to the manuscript are acceptable to the reviewers and we thank them again for their efforts in helping us improve the manuscript.